# DBQ-SSD: Dynamic Ball Query for Efficient 3D Object Detection

**Jinrong Yang**[1][*]  **Lin Song**[2][*]  **Songtao Liu**[3]  **Weixin Mao**[3]
**Zeming Li**[3]  **Xiaoping Li**[1][†]  **Hongbin Sun**[4][†]  **Jian Sun**[3]  **Nanning Zheng**[4]
[1]Huazhong University of Science and Technology
[2]Tencent AI Lab       [3]MEGVII Technology       [4]Xi'an Jiaotong University
`yangjinrong@hust.edu.cn, ronnysong@tencent.com`

## Abstract

Many point-based 3D detectors adopt point-feature sampling strategies to drop some points for efficient inference. These strategies are typically based on fixed and handcrafted rules, making it difficult to handle complicated scenes. Different from them, we propose a Dynamic Ball Query (DBQ) network to adaptively select a subset of input points according to the input features, and assign the feature transform with a suitable receptive field for each selected point. It can be embedded into some state-of-the-art 3D detectors and trained in an end-to-end manner, which significantly reduces the computational cost. Extensive experiments demonstrate that our method can increase the inference speed by 30%-100% on KITTI, Waymo, and ONCE datasets. Specifically, the inference speed of our detector can reach 162 FPS on KITTI scene, and 30 FPS on Waymo and ONCE scenes without performance degradation. Due to skipping the redundant points, some evaluation metrics show significant improvements.

## 1 Introduction

3D object detection, as a fundamental task in the 3D scene, has made significant progress. It aims to recognize and localize objects from point clouds and paves the way for several real applications like autonomous driving (Geiger et al., 2012), robotic system(Yang et al., 2020b), and augmented reality (Park et al., 2008).

The structure of point clouds is sparse, unordered, and semantically deficient, making it difficult to encode point features like highly structured image features. To eliminate this barrier, voxel-based methods are proposed to organize the overall point cloud as neatly distributed voxels. Therefore, the naive 3D convolution or its efficient variant, *i.e.*, 3D sparse convolution (Yan et al., 2018), can be used to extract voxel features like image processing manner. Although the voxel-based methods bring convenience to the processing of point cloud, they are prone to drop detail information, making it inescapable to suffer from suboptimal performance. Another stream of methods are point-based methods (Yang et al., 2020b; Chen et al., 2022; Zhang et al., 2022) inspired from PointNet++ (Qi et al., 2017). They employ a series of operations, *i.e.*, farthest point sample (FPS), query, and grouping, to directly extract informative features from the naive point cloud. However, the straightforward pipeline is cumbersome and costly. 3D-SSD (Yang et al., 2020b) first propose a single stage architecture, *i.e.*, using encoder-only architecture, which replaces the geometric Distance-based FPS (D-FPS) with the Feature similarity-based FPS (F-FPS) to recall more foreground point features and further discard feature propagation (FP) layers for reducing latency. Although eliminating the overhead of FP layer, F-FPS operations still occupy vast latency. IA-SSD (Zhang et al., 2022) further proposes a contextual centroid prediction module to replace F-FPS, which directly predicts the classification scores of each point and adopts efficient top-k operation to further recall more foreground, and further cut computation overhead.

The current advanced designs mainly credit to efficient foreground recall, but they may still have redundancy in the other parts like the background points or the network structure. In this paper, we

---

[*]Equal contribution.
[†]Corresponding author.

conduct several empirical analyses of IA-SSD (Zhang et al., 2022) on two representative benchmarks, *i.e.*, KITTI (Geiger et al., 2012) and Waymo (Sun et al., 2020), to uncover the full picture of its inference speed bottleneck. We first calculate the latency distribution of all detector modules to figure out which parts are the main speed bottlenecks. Then we count the ratio of background points and the scale distribution of all objects to further explore potential redundancy clues over the cumbersome modules. Our study reveals three valuable points: (1) MLP network occupies over half latency. (2) Tremendous spatial redundancy exists in background point features that appear in each stage of the detector. (3) The size of each object is varying, making it unusable to align each receptive field of the conventional multi-scale grouping (MSG) and suffering from branch redundancy in MSG.

The above valuable finding motivates us to further build a more efficient detector with higher speed by reducing the redundant background points and blocking useless MSG branches. As shown in Fig. 2, we propose Dynamic Ball Query (DBQ) to replace the vanilla ball querying module, where the vanilla ball querying means the sampling technique proposed by PointNet++ (Qi et al., 2017). It dynamically activates useful and compact point features, and blocks redundant background point features for each branch of MSG. For each point feature sampled by FPS or top-k classification score, we design a dynamic query multiplexer to determine which branch to go through. Specifically, we apply a light-weight MLP network for point features to predict N masks corresponding to N branches of MSG, where the value of the mask is $\{0, 1\}$ corresponding to blocking and activating states. The overall dynamic router procedure is data-driven so that the point features are adaptively activated or blocked with a suitable combination among all receptive fields of MSG. Ultimately, we introduce a resource budget loss for DBQ to learn a trade-off between latency and performance.

To verify the efficiency of our method, we conduct extensive experiments on three typical datasets, *i.e.*, KITTI (Geiger et al., 2012), Waymo (Sun et al., 2020), ONCE (Mao et al., 2021b). Our Dynamic Ball Query enables the 3D detector to cut the latency from 9.85 ms (102 FPS) to 6.172 ms (162 FPS) on KITTI scene and speed up the inference speed from 20 FPS to 30 FPS on Waymo and ONCE scene while keeping the comparable performance. In particular, some evaluation metrics show significant improvements.

## 2 RELATED WORK

### 2.1 3D DETECTORS

The task of 3D detection is to predict 3D bounding boxes and class labels for each object in a point cloud scene. The detection algorithms can be split into voxel-based (Zhou & Tuzel, 2018; Yan et al., 2018; Lang et al., 2019; He et al., 2020) and point-based (Shi et al., 2019; Yang et al., 2020b; Chen et al., 2022; Zhang et al., 2022) methods. Voxel-based methods convert the point cloud into regular voxels or pillars, making it natural to apply 3D convolution or its sparse variant (Yan et al., 2018) for feature extraction. This regular partition method may lead to detailed information lost, so point-based methods are proposed to directly process vanilla point cloud. Inspired by PointNet++ (Qi et al., 2017) and Faster R-CNN (Ren et al., 2015), PointRCNN (Shi et al., 2019) employs SA and FP layers to extract feature for each point and designs a region proposal network (RPN) to produce proposals, and utilizes an extra stage of the module to predict bounding boxes and class labels. In addition, PV-RCNN (Shi et al., 2020a) integrates voxel and point features to the RPN for generating higher-quality proposals. Pyramid R-CNN (Mao et al., 2021a) introduces a pyramid RoI head with learnable radii to boost accuracy at the expense of latency overhead. In contrast, our method aims to achieve more efficient inference by adaptively selecting the network branches for each input point. Other efforts are similar to single-stage 2D detectors (Lin et al., 2017; Tian et al., 2019; Wang et al., 2021; Song et al., 2019b; Zhang et al., 2019). 3DSSD (Yang et al., 2020b) and IA-SSD (Zhang et al., 2022) discard the region proposal network and use encoder-only architecture to localize 3D objects. Our work focus on dynamically dropping the redundant background point features for single-stage point-based detector, which is rarely researched in previous works. Our method endows the detector with super inference speed.

### 2.2 EFFICIENT 3D POINT-BASED DETECTORS

The point-based methods need to process large-scale vanilla point features, which requires to build cumbersome models and suffers expensive computation costs. Therefore, several works (Yang et al.,

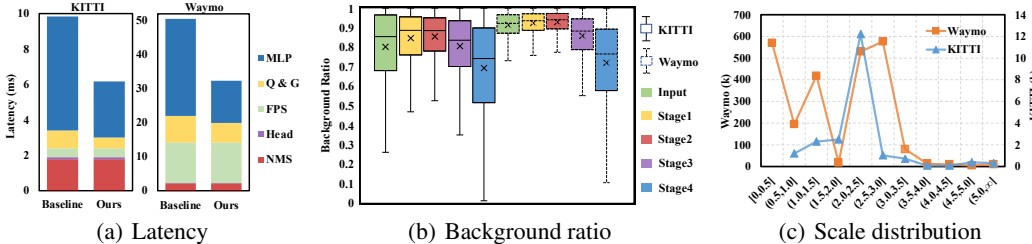

(a) Latency          (b) Background ratio          (c) Scale distribution

Figure 1: Statistics of latency, background ratio, and size distribution on both KITTI *val* (Geiger et al., 2012) and Waymo *val* (Sun et al., 2020) sets. (a) reveals that the MLP network occupies the largest latency. "Q & G" means query and grouping operation. (b) reflects that redundant background points significantly dominate the input points of each stage. (c) means the distribution on varying object sizes (measuring in $\sqrt[3]{volume}$, where $volume$ is the volume of ground truth).

2020b; Chen et al., 2022; Zhang et al., 2022) aim at designing a lightweight and effective point-based detector. 3DSSD (Yang et al., 2020b) proposes a feature-based fastest point sampling (F-FPS) strategy by measuring the similarity of point features to replace the geometric distance of D-FPS (Qi et al., 2017). The policy makes it reasonable to discard FP layers since F-FPS can recall more foreground points instead of compensating for foreground information by cumbersome FP layers. Even so, the F-FPS operation suffers from high complexity bottleneck. IA-SSD introduces a contextual centroid prediction module for replacing the F-FPS operation to efficiently recall more foreground points. It predicts the classification score for each point feature and applies the efficient top-k operation to all points ordered by category independent scores. Instead of focusing on the foreground part by efficient sampling policies, we explore the spatial redundancy of background points. We introduce a dynamic network mechanism to adaptively reduce useless background points in a data-dependent regime.

## 2.3 Dynamic Network

Dynamic networks target to adaptively change the structure of networks and parameters in a data-driven manner. In the field of model automation designing, dynamic network mechanism is used to drop blocks (Wu et al., 2018; Huang et al., 2017; Mullapudi et al., 2018; Wang et al., 2018), pruning channels (Lin et al., 2018; You et al., 2019; Jiang et al., 2018), adjusting layer-level scales (Li et al., 2020b; Yang et al., 2020a) and changing spatial typologies (Song et al., 2019a; 2020b). DRNet (Li et al., 2020b) aims at learning an automatic scale transformation for a feature pyramid network in the semantic segmentation scene. Switch Transformer (Fedus et al., 2021) employs the Mixture of Experts (MoE) (Shazeer et al., 2017) model to choose different parameters for each input data. Deformable convolution (Dai et al., 2017) learns a convolution of arbitrary shapes by predicting an offset for each parameter. Dynamic Grained Encoder (Song et al., 2021) adopts a dynamic gate mechanism to select the spatial granularity of input query in the transformer network. Dynamic Convolution (Verelst & Tuytelaars, 2020), Dynamic Head (Song et al., 2020a) focus on learning a sparse mask for network to discard redundant image features. In this paper, we focus on 3D detection on point cloud scenes and aim at designing a point-wise dynamic network to drop redundant background points.

## 3 Analysis and Motivation

To explore what hampers the higher speed for 3D detection task, we conduct several experiments on both KITTI *val* and Waymo *val* sets with the state-of-the-art point-based 3D-detector (Zhang et al., 2022). To establish a strong baseline, we split the point cloud into four parallel parts to speed up the first FPS operation. We first measure the *latency* on each detector module. As shown in Fig. 1(a), the MLP network occupies the major part of the overall time overhead, *i.e.,* 6.44 ms (65.4%) and 28.56 ms (56.5%) on KITTI and Waymo scenes, respectively. Therefore, optimizing the cumbersome and costly MLP network is a top priority for building an efficient detector.

Decreasing the input scale of points or network parameters is a natural and empirical manner for reducing the latency but the policies are prone to damage the performance. To avoid both notorious drawbacks, we attempt to analyze the redundancy of background points in each backbone stage. As

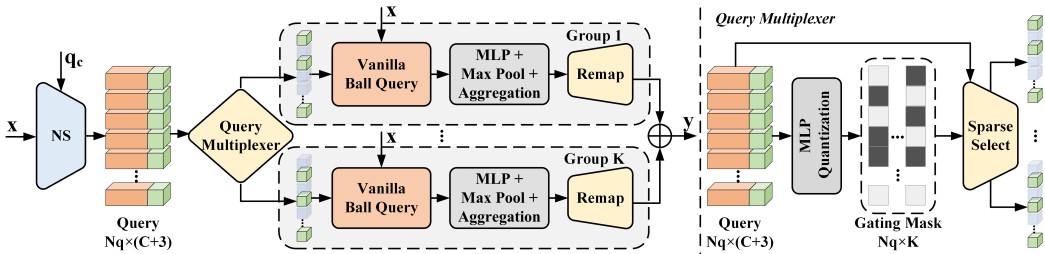

Figure 2: The pipeline of dynamic ball query in a set abstraction layer. 'NS' indicates the nearest sampling, which samples the query features from the input features. The query multiplexer generates gating masks to adaptively select a subset of input queries for each group. The remap operator is used to map the sparse features to the dense form.

shown in Fig. 1(b), it indicates that the number of background points (point features) dominates the proportion of over 70% at any network level. The phenomenon reveals that significant redundancy exists in background points which may be discarded for speeding up the detection procedure.

Going one step further, we point out that the conventional multi-scale grouping (MSG) operation (Qi et al., 2017) of the set abstraction (SA) layer may be also redundant. As shown in Fig. 1(c), it reflects that the size of each object varies in either KITTI or Waymo scenes. Therefore, the given receptive field of MSG may not entirely align with the size of objects. In this regime, some grouping branches of MSG are useless. The valuable observation motivates us to choose optimal grouping branches, which can further make the detector efficient.

## 4 DYNAMIC BALL QUERYING

Our DBQ-SSD framework is established on the efficient IA-SSD framework, which adopts set abstraction (SA) layers to encode point features. To achieve a better balance between effectiveness and efficiency, we introduce the dynamic network mechanism into the IA-SSD framework. Specifically, we propose Dynamic Ball Querying (DBQ) to replace the vanilla ball querying in each set of abstraction layer, which is shown in Fig. 2. It is able to adaptively select a subset of input points as queries and extract local features in a suitable spatial radius. Dynamic ball querying is a generic module which can be easily embedded into the encoder blocks in mainstream point-based 3D detection frameworks.

Given an input point sequence $\mathbf{x_c} \in \mathbb{R}^{N \times 3}$ with the corresponding features $\mathbf{x_f} \in \mathbb{R}^{N \times C}$, $N$ denotes the length of sequence and $C$ indicates the number of feature channels. Besides, the coordinate sequence of input queries is denoted as $\mathbf{q_c} \in \mathbb{R}^{N_q \times 3}$, where $N_q$ is the number of queries. For efficiency, most previous point-based methods (Yang et al., 2020b; Zhang et al., 2022) use Farthest Point Sampling (FPS) or its variants (*e.g.,* dividing point cloud by multiple parallel parts to reduce computational complexity) to generate the query points. For each query, the vanilla ball querying samples a predefined number of local points within a specific spatial radius. Since different object instances require different receptive fields, in a set abstraction layer, the previous 3D detectors (Yang et al., 2020b; Zhang et al., 2022) typically adopt multiple groups of vanilla ball querying with different radii to increase feature diversity.

Specifically, for one set abstraction layer, we define the set of predefined radii as $R = \{r_k\}^K$ and the number of sampling points as $\Phi = \{\phi_k\}^K$, where $K$ indicates the number of groups. Based on these, we establish a set of $K$ vanilla ball querying blocks. As shown in Fig. 2, our dynamic ball querying is made up of a query multiplexer and a set of vanilla ball querying blocks. Similar to the gate-based dynamic networks (Song et al., 2021; Wang et al., 2018; Li et al., 2020b), the query multiplexer adopts a fine-grained routing process to select a suitable combination of vanilla ball querying blocks for each query.

### 4.1 INFERENCE

For an input query $i \in \{1, 2, ..., N_q\}$ with coordinate $\mathbf{q_c}(i)$, we first obtain the corresponding query feature according to its coordinate for the query multiplexer. The query feature can be used as the

basis for dynamic decisions. Specifically, we adopt the nearest sampling technique to obtain its representation from the features of input points. Albeit the sampling process with more sample points, *e.g.*, top-k sampling, and ball gathering, can lead to a slight performance gain, it causes a significant decrease in efficiency. The top-k sampling indicates choosing a set of points with the smallest distance under a specific metric. Additionally, ball gathering means sampling and concatenating features according to the coordinates of points. The previous variants of ball querying employ heuristic and hand-designed rules. Different from them, the routing process of our query multiplexer is performed in a data-dependent way. To achieve it, we aggregate input features for each query from the nearest input point and predict the gating logits for each query by a linear projection:

$$\mathbf{h}(i) = \mathbf{x_f}(\arg\min_j \|\mathbf{q_c}(i) - \mathbf{x_c}(j)\|)\mathbf{W} + b \in \mathbb{R}^{1 \times K}, \tag{1}$$

where $\mathbf{W} \in \mathbb{R}^{C \times K}$ and $b \in \mathbb{R}^{1 \times K}$ denotes the weight and bias, respectively. Moreover, the binary gating masks for the $i$-th query and the $k$-th group are generated by quantizing the gating logits:

$$\mathbf{m}(i,k) = \text{step}(\mathbf{h}(i,k)), \text{ where step}(\mathbf{h}(i,k)) = \left\{ \begin{array}{ll} 1, & \text{if } \mathbf{h}(i,k) \geq 0 \\ 0, & \text{if } otherwise \end{array} \right. \tag{2}$$

The gating masks control which ball querying group is enabled, *i.e.*, the group with a positive mask value is enabled and vice versa. Based on this, we can adaptively reduce the number of queries and obtain the coordinates of sparse queries for each group:

$$\hat{\mathbf{q_c}}(k) = \{\mathbf{q_c}(i)|\mathbf{m}(i,k) \neq 0, \forall i \in 1, 2, ..., N_q\}, \forall k \in \{1, 2, ..., K\}, \tag{3}$$

Furthermore, the sparse coordinates are used to guide the vanilla ball querying with corresponding settings to generate the sparse query features. Following the conventional protocols in the PointNet-like methods, the sparse query features are then transformed by the predefined MLP layers and max pooling operator:

$$\hat{\mathbf{z}}(k) = \text{MaxPool}(\text{MLP}_k(\text{VanillaBallQuery}(\hat{\mathbf{q_c}}(k); \mathbf{x_c}, \mathbf{x_f}, r_k, \phi_k))), \tag{4}$$

To fuse the sparse transformed features in different groups, we remap them into the dense form according to the gating mask. The remap operator is similar to the unpooling process, which projects each enabled feature to the original position and fills zero to the disabled positions. The output feature of the SA layer are then blended by summation:

$$\mathbf{y_f} = \sum_{k \in \{1,...,K\}} \mathbf{z}(k), \text{ where } \mathbf{z}(k) = \text{Aggregation}_k(\text{Remap}(\hat{\mathbf{z}}(k); \mathbf{h})), \tag{5}$$

where the Aggregation$_k$ is a linear layer, which is used to transform the features in different groups into the same dimension.

## 4.2 TRAINING

Since the sparse selection in Eq. 3 is non-differentiable, it is nontrivial for the dynamic ball querying to enable fully end-to-end training. To achieve it, we replace the determined decisions in Eq. 3 with a stochastic sampling process. Concretely, we consider the gating logits unnormalized log probabilities under the Bernoulli distribution. To this end, with noise samples $g$ and $g'$ drawn from a standard Gumbel distribution (Gumbel, 1954), a discrete gating mask can be yielded:

$$\mathbf{m}(i,k) = \text{step}(\mathbf{h}(i,k) + g - g'), \text{ where } g, g' \sim \text{Gumbel}(0,1). \tag{6}$$

To enable end-to-end training, motivated by the previous dynamic networks, we use the Gumbel-Sigmoid technique (Gumbel, 1954) to give a differentiable approximation for the Eq. 6 by replacing the hard step function with the soft sigmoid function. The likelihood of the $i$-th query in $k$-th group being selected is:

$$\pi(i,k) = \frac{\exp((\mathbf{h}(i,k) + g)/\tau)}{\exp((\mathbf{h}(i,k) + g)/\tau) + \exp(g'/\tau)} \in [0,1], \tag{7}$$

where $\tau$ is the temperature coefficient. In the training phase, we use the Eq. 6 as the gating mask to select the sparse queries and employ a straight-through estimator (Bengio et al., 2013; Verelst & Tuytelaars, 2020) to obtain the gradients of gating logits:

$$\mathbf{y_f}(i) = \left\{ \begin{array}{ll} \sum_k^K \mathbf{z}(i,k) & \text{forward} \\ \sum_k^K \pi(i,k) \cdot \mathbf{z}(i,k) & \text{backward} \end{array} \right. \tag{8}$$

### 4.3 LATENCY CONSTRAINT

Without latency constraint, dynamic ball querying typically enables more queries for each group to obtain high accuracy. To achieve a better balance between effectiveness and efficiency, we introduce the latency constraint as a training target to reduce the inference time. Different from the computational complexity employed in many previous dynamic networks (Song et al., 2021; Wang et al., 2018; Li et al., 2020b), the latency can represent the actual runtime in specific devices. To this end, we first establish a latency map for each group in each SA layer, which records the latency with regard to the number of queries. Based on this, we can calculate the latency ratio of all the SA layers with dynamic ball querying:

$$\Psi = \frac{\sum_l \sum_k \Psi_{l,k}(\sum_i \mathbf{m}^l(i,k)))}{\sum_l \sum_k \Psi_{l,k}(N_q^l)}, \tag{9}$$

where $\Psi_{l,k}$ indicates the latency map of $k$-th group in $l$-th layer. Finally, the latency is constrained by using the Euclidean distance to design a budget loss, thus the total loss is:

$$\mathcal{L} = \mathcal{L}_{\text{tasks}} + \lambda \mathcal{L}_{\text{budget}}, \text{ where } \mathcal{L}_{\text{budget}} = |\Psi - \gamma|. \tag{10}$$

For simplicity, the objective latency budget $\gamma$ is set to 0 in all experiments. The hyper-parameter $\lambda$ is used to scale the budget loss. In addition, if the input is batched format, $\Psi$ needs to be averaged along the batch dimension to estimate the average overhead of the network.

## 5 EXPERIMENT

### 5.1 SETTING

**Dataset** We evaluate our detector on two representative datasets: KITTI dataset (Geiger et al., 2012) and Waymo dataset (Sun et al., 2020). KITTI dataset includes 7,481 training point clouds/images and 7,518 test point clouds/images. The KITTI scene contains three classes, *i.e.*, *Car*, *Pedestrian*, and *Cyclist*. Waymo scene contains 798 training, 202 validation, and 150 testing sequences with three classes of *Vehicle*, *Pedestrian*, and *Cyclist*. Each sequence includes nearly 200 frames with a 360-degree lidar point cloud.

**Evaluation metrics.** For KITTI scene, we report the performance of all classes by measuring the average precision (AP) metric. Follow most of state-of-the-art methods, we adopt 0.7, 0.5, and 0.5 of the IoU thresholds for *Car*, *Pedestrian*, and *Cyclist*, respectively. In addition, three levels of difficulty ("Easy", "Moderate", and "Hard") are also reported. To evaluate Waymo, we use the official metrics, Average Precision (AP) and Average Precision weighted by Heading (APH), and report the performance on LEVEL 1 (L1) and LEVEL 2 (L2) difficulty levels.

**Implementation Details** Following the stream of single-stage point-based methods, we use encoder-only architecture like (Yang et al., 2020b; Chen et al., 2022; Zhang et al., 2022). Specially, we split the point features into four parallel fan parts to speed up D-FPS of the first sampling layer, which will be acted as the "Efficient Baseline". Other sampling layers follow the default setting of (Zhang et al., 2022). We employ two groups of each MSG with different radius ([0.2, 0.8], [0.8, 1.6], [1.6, 4.8], [4.8, 6.4]) to aggregate point features. We set the temperature $\tau = 1$ for all the experiments. All experiments are implemented by OpenPCDet [1] framework.

### 5.2 EVALUATION ON KITTI DATASET

We randomly sample 16,384 points from the overall point cloud per single view frame. We train our model by ADAM (Kingma & Ba, 2014) optimizer with onecycle learning strategy (Smith & Topin, 2019). The batch size is set to 16 with 8 GPUs. The initial learning rate is 0.01 and is decayed by 0.1 at 35 and 45 epochs.

---

[1]https://github.com/open-mmlab/OpenPCDet

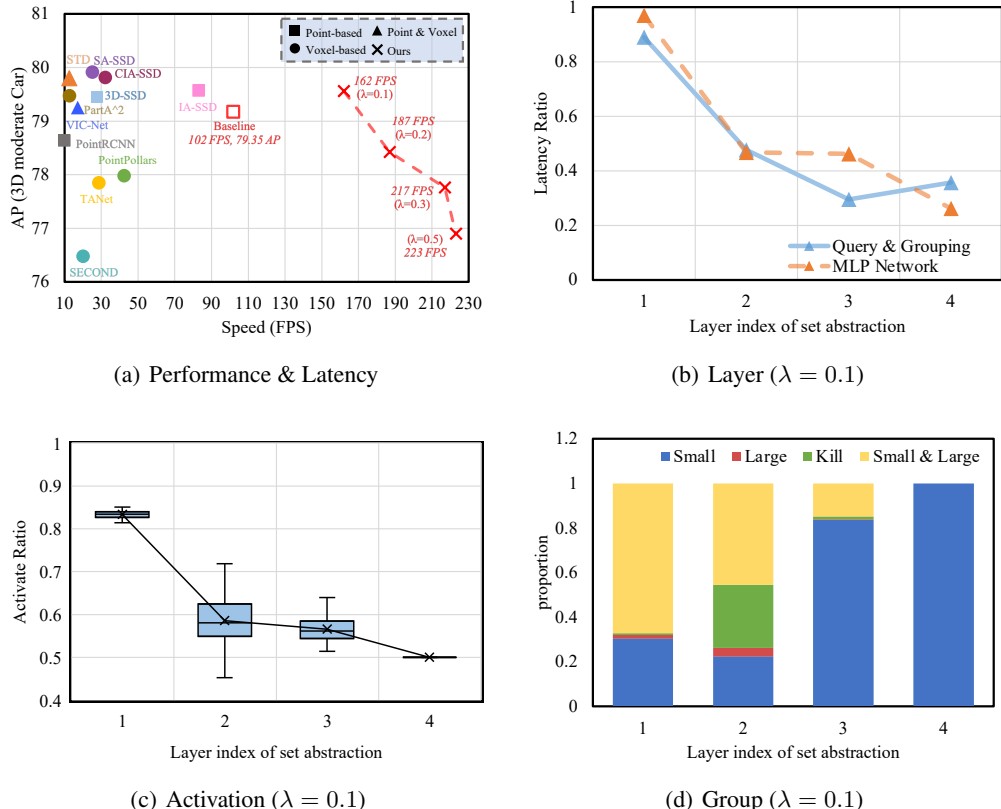

Figure 3: Illustration of the effects on Dynamic Ball Query. All experiments are evaluated on KITTI *val* set. $\lambda$ is the scale parameter of resource budget loss in Eq. 10. Latency here is evaluated by a single RTX2080Ti GPU with a batch size of 16. (a) reports the comparison on both accuracies of *Car* class and overall latency distribution. (b) indicates the latency reduction of query & grouping operation and MLP network in different SA layers. (c) reflects the activation distribution of point features in different SA layers. (d) shows the proportion of point features go through different groups of MSG. "Small" and "Large" means activating on group branches with small and large radii respectively. "Kill" represents blocking all groups, while "Small & Large" means going through all scales of groups.

Table 1: Performance of dynamic gating with different routing manners on KITTI *val* set. The scale parameter $\lambda$ is set to 0.1. "Layer" indicates controlling an entire SA layer. "Share" means whether to share masks to all groups. "Point" indicates using point-wise routing instead of layer-wise routing.

| Dynamic | Routing | | | 3D Car (IoU=0.7) | | | 3D Pedestrian (IoU=0.5) | | | 3D Cyclist (IoU=0.5) | | | Latency |
| | Layer | Share | Point | Easy | Moderate | Hard | Easy | Moderate | Hard | Easy | Moderate | Hard | (FPS) |
|---|---|---|---|---|---|---|---|---|---|---|---|---|---|
| ✗ | - | - | - | 88.89 | 79.17 | 77.96 | 61.01 | 58.11 | 52.62 | 84.83 | 69.29 | 65.72 | 102 |
| ✓ | ✓ | | | 89.19 | 79.33 | 78.31 | 61.32 | 58.60 | 52.75 | 85.81 | 70.77 | 66.04 | 120 |
| | | ✓ | ✓ | 87.28 | 77.34 | 76.29 | 60.43 | 56.79 | 51.45 | 84.83 | 68.88 | 65.12 | **165** |
| | | | ✓ | **89.60** | **79.56** | **78.51** | **61.57** | **58.82** | **53.11** | **85.92** | **71.03** | **66.33** | 162 |

**Dynamic vs Static**  To verify the efficiency of our method, we first report the performance and latency on KITTI scene. As shown in Fig. 3(a), our detector achieves super speed while maintaining comparable performance with other detectors. When $\lambda$ is set to 0.1, the performance surpasses the efficient baseline. Therefore, we set the scale parameter to 0.1 for all experiments by default. As increasing the supervision of resource budget loss, the speed is further improved to 223 FPS. The impressive results show that our method endows point-based 3D detector with efficient detection capability. Going one step further, we report the reduction of latency on two revenue modules, *i.e.*, query & grouping operation, and MLP network. As shown in Fig. 3(b), our method cut the considerable overhead of both in different SA layers. It verifies that adaptively turning off useless points by DBQ can speed up the computation of both modules.

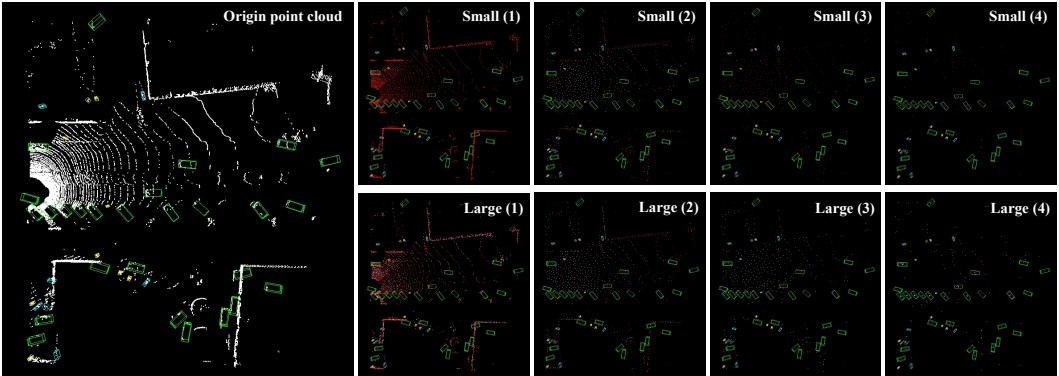

Figure 4: Visualization results on KITTI *val* set. The 3D boxes in the figures are the prediction boxes. Green, cyan, and yellow represent *Car, Pedestrian, and Cyclist*. Red and white points represent activation and blocking points, respectively. "Small" and "Large" means the scale of group in MSG, and the digital in parentheses is the index of SA layer.

**Activation and Blocking Points** To explore which points are activated or blocked, we conduct quantitative and qualitative experiments using visualization and statistics respectively. Fig. 3(c) counts the activation ratios of point features in different layers. It indicates that a considerable large part of points is discarded. Next, we dip into visualization results to figure out the reason for the quantitative results. As shown in Fig. 4, the deeper the network goes, the blocking point ratio is larger. This is because it learns rich semantic information to judge useful foreground and redundant background points when going deeper into the network. As for the early layer, high-level semantic knowledge is difficult to extract. Therefore, the model is hard to make decisions to block out background points. In general, it reflects clearly that the most activating points (red points) in the deeper layer are distributed around objects, while other points away from the objects are blocked. It reveals that our method discards most redundant background points which may be useless for the localization and classification of objects, making it reasonable for the detector to speed up inference. The remaining foreground points and surrounding points are used to support the structure of objects and enrich context information. Therefore, our method does not damage the performance of detection. It echoes our analysis in Sec. 3.

**Routing Manner and Branch Redundancy** Tab. 1 compares the effectiveness between layer-wise and point-wise routing manner. During inference, the layer-wise routing manner only speeds up by 18 FPS with small performance gains, while our point-wise manner not only achieves significant performance improvement but also reduces considerable latency. In addition, the contrast on whether to predict split mask or share identity mask for each group of MSG indicates that the former is the optimal policy. Generating two masks for different scales of group allow each point to reach an optimal combination of the receptive field. Corresponding statistical and visualization results can be seen in Fig. 3(d) and Fig. 4, which reveals that more points go through both groups in early layers while most points are only keen on a single branch or even be killed. The phenomenon agrees well with the empirical analysis in Sec. 3.

**Main Results** As illustrated in Tab. 2, our method outperforms the efficient baseline on all categories, while achieving higher inference speed (162 vs 102). It verifies that our method can not only drop redundant points for speeding up inference, but extract more useful information for localization and classification. By comparing with other state-of-the-art methods, our detector outperforms their speed with a large margin while maintains comparable accuracy.

### 5.3 EVALUATION ON WAYMO DATASET

To verify the generalization of our method, we further evaluate the performance on Waymo (Sun et al., 2020) dataset. Because Waymo scene is made up of the 360-degree point cloud whose scale is larger than KITTI scene, we increase the input number of points by sampling from 16,384 to 65,536. The batch size is set to 2 for each GPU. We train 30 epochs with 8 GPUs. Other settings are the same as the experiments of KITTI scene.

Table 2: Comparison with the state-of-the-art methods on the KITTßI *test* set. Bold font is used to indicate the best performance. The speed is tested on a single GPU with with batch size of 16 and measured by FPS.

| Method | Type | 3D Car (IoU=0.7) | | | 3D Ped. (IoU=0.5) | | | 3D Cyc. (IoU=0.5) | | | Speed |
|---|---|---|---|---|---|---|---|---|---|---|---|
| | | Easy | Mod. | Hard | Easy | Mod. | Hard | Easy | Mod. | Hard | |
| Voxel-based Methods | | | | | | | | | | | |
| VoxelNet (Zhou & Tuzel, 2018) | 1-stage | 77.47 | 65.11 | 57.73 | 39.48 | 33.69 | 31.5 | 61.22 | 48.36 | 44.37 | 4.5 |
| SECOND (Yan et al., 2018) | 1-stage | 84.65 | 75.96 | 68.71 | 45.31 | 35.52 | 33.14 | 75.83 | 60.82 | 53.67 | 20 |
| PointPillars (Lang et al., 2019) | 1-stage | 82.58 | 74.31 | 68.99 | 51.45 | 41.92 | 38.89 | 77.10 | 58.65 | 51.92 | 42.4 |
| 3D IoU Loss (Zhou et al., 2019) | 1-stage | 86.16 | 76.50 | 71.39 | - | - | - | - | - | - | 12.5 |
| Associate-3Ddet (Du et al., 2020) | 1-stage | 85.99 | 77.40 | 70.53 | - | - | - | - | - | - | 20 |
| SA-SSD He et al. (2020) | 1-stage | 88.75 | 79.79 | 74.16 | - | - | - | - | - | - | 25 |
| CIA-SSD (Zheng et al., 2020) | 1-stage | 89.59 | 80.28 | 72.87 | - | - | - | - | - | - | 32 |
| TANet Liu et al. (2020) | 2-stage | 84.39 | 75.94 | 68.82 | 53.72 | **44.34** | 40.49 | 75.70 | 59.44 | 52.53 | 28.5 |
| Part-A$^2$ | 2-stage | 87.81 | 78.49 | 73.51 | 53.10 | 43.35 | 40.06 | **79.17** | 63.52 | 56.93 | 12.5 |
| Point-Voxel Methods | | | | | | | | | | | |
| Fast Point R-CNN Chen et al. (2019) | 2-stage | 89.29 | 77.40 | 70.24 | - | - | - | - | - | - | 16.7 |
| STD (Yang et al., 2019) | 2-stage | 87.95 | 79.71 | 75.09 | 53.29 | 42.47 | 38.35 | 78.69 | 61.59 | 55.30 | 12.5 |
| PV-RCNN (Shi et al., 2020a) | 1-stage | **90.25** | **81.43** | **76.82** | 52.17 | 43.29 | 40.29 | 78.60 | 63.71 | **57.65** | 12.5 |
| VIC-Net (Jiang et al., 2021) | 1-stage | 88.25 | 80.61 | 75.83 | 43.82 | 37.18 | 35.35 | 78.29 | 63.65 | 57.27 | 17 |
| HVPR (Noh et al., 2021) | 1-stage | 86.38 | 77.92 | 73.04 | 52.47 | 43.96 | **40.64** | - | - | - | 36.1 |
| Point-based Methods | | | | | | | | | | | |
| PointRCNN (Shi et al., 2019) | 2-stage | 86.96 | 75.64 | 70.70 | 47.98 | 39.37 | 36.01 | 74.96 | 58.82 | 52.53 | 10 |
| 3D IoU-Net (Li et al., 2020a) | 2-stage | 87.96 | 79.03 | 72.78 | - | - | - | - | - | - | 10 |
| Point-GNN (Shi & Rajkumar, 2020) | 1-stage | 88.33 | 79.47 | 72.29 | 51.92 | 43.77 | 40.14 | 78.60 | 63.48 | 57.08 | 1.6 |
| 3DSSD (Yang et al., 2020b) | 1-stage | 88.36 | 79.57 | 74.55 | **54.64** | 44.27 | 40.23 | 82.48 | **64.10** | 56.90 | 25 |
| IA-SSD (Zhang et al., 2022) | 1-stage | 88.34 | 80.13 | 75.04 | 46.51 | 39.03 | 35.60 | 78.35 | 61.94 | 55.70 | 83 |
| IA-SSD (Reproduced) | 1-stage | 87.67 | 79.40 | 74.22 | 46.16 | 38.29 | 35.61 | 78.26 | 61.53 | 55.48 | 83 |
| DBQ-SSD | 1-stage | 87.93 | 79.39 | 74.40 | 47.59 | 38.08 | 35.61 | 78.18 | 62.80 | 55.70 | **162** |

Table 3: Comparison with the state-of-the-art methods on the Waymo *val* set. The bold font is used to indicate best performance. The speed is tested on a single GPU with batch size of 16 and measured by FPS.

| Method | Vehicle (LEVEL 1) | | Vehicle (LEVEL 2) | | Ped. (LEVEL 1) | | Ped. (LEVEL 2) | | Cyc. (LEVEL 1) | | Cyc. (LEVEL 2) | | speed |
|---|---|---|---|---|---|---|---|---|---|---|---|---|---|
| | mAP | mAPH | mAP | mAPH | mAP | mAPH | mAP | mAPH | mAP | mAPH | mAP | mAPH | |
| PointPollars (Lang et al., 2019) | 60.67 | 59.79 | 52.78 | 52.01 | 43.49 | 23.51 | 37.32 | 20.17 | 35.94 | 28.34 | 34.60 | 27.29 | - |
| SECOND (Yan et al., 2018) | 68.03 | 67.44 | 59.57 | 59.04 | 61.14 | 50.33 | 53.00 | 43.56 | 54.66 | 53.31 | 52.67 | 51.37 | - |
| Part-A$^2$ (Shi et al., 2020b) | 71.82 | 71.29 | 64.33 | 63.82 | 63.15 | 54.96 | 54.24 | 47.11 | 65.23 | 63.92 | 62.61 | 61.35 | - |
| PV-RCNN (Shi et al., 2020a) | **74.06** | **73.38** | **64.99** | **64.38** | 62.66 | 52.68 | 53.80 | 45.14 | 63.32 | 61.71 | 60.72 | 59.18 | - |
| IA-SSD (Zhang et al., 2022) | 70.53 | 69.67 | 61.55 | 60.80 | **69.38** | 58.47 | **60.30** | 50.73 | 67.67 | 65.30 | 64.98 | 62.71 | 14 |
| Efficient Baseline | 71.15 | 70.30 | 62.49 | 61.73 | 68.38 | 58.21 | 59.75 | 50.80 | **68.64** | **66.23** | **66.09** | 63.78 | 20 |
| DBQ-SSD ($\lambda$=0.1) | 70.56 | 69.82 | 61.81 | 61.15 | 68.89 | 58.07 | 60.15 | 50.60 | 66.58 | 63.98 | 64.22 | 61.66 | **30** |
| DBQ-SSD ($\lambda$=0.05) | 71.58 | 71.03 | 64.13 | 63.61 | 69.18 | **58.47** | 60.22 | **50.81** | 68.29 | 66.01 | **66.09** | **63.86** | 27 |

As shown in Tab 3, we report the performance and inference speed of our DBQ-SSD. As carrying out suitable supervision (*i.e.*, $\lambda = 0.05$) for the detector, it achieves impressive performance in all classes. Especially, some categories achieve state-of-the-art accuracy in evaluation metrics or levels of difficulty. Meanwhile, it also improves the inference speed of the efficient baseline from 20 FPS to 27 FPS. The results show that adaptively blocking redundant point features and activating high-quality point features are the key to endow our detector with efficient performance. When conducting a larger scale of supervision for DBQ-SSD, it is capable of detecting objects with real-time speed (30 FPS) and achieves comparable accuracy. It can provide flexible configuration for practical applications to realize the best trade-off between accuracy and overhead cost.

## 6 CONCLUSION

In this paper, we point out the existing spatial redundancy on background points and useless receptive field groups in MSG. This redundancy impedes the inference efficiency improvement of point-based 3D detectors. To eliminate the dilemma, we propose a dynamic ball query, which can dynamically generate gate masks for each group of MSG to process useful points and block redundant background points. The extensive experiments demonstrate our analysis and show the effectiveness of our method. In short, we launch a new view to focus on redundant background points instead of the limited foreground part, which further deepens the understanding of the sparsity of point cloud. We hope this work can shed the light to the research of efficient point-based models and inspire future works.

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

## A    LIMITATION AND FUTURE WORK

With increasing the supervision on resource budget (increasing the value of $\lambda$.), the performance will decrease accordingly. We suspect that dropping too many point clouds may eliminate the part of the useful point cloud. Therefore, this paper targets to achieve a better trade-off between accuracy and inference speed, which maintaining or even achieving gain in accuracy, and significantly speeding up inference. In this paper, we can not specifying what point to drop but our detector equips strong ability to eliminate redundancy. Therefore, we look forward to inspire future works for focusing on dropping more redundant point cloud without performance degradation.

Table 4: Illustration of the architecture of DBQ-SSD. npoint denotes the number of sampled points, [radii] denotes the grouping radii, [nquery] denotes the number of grouping points, [dimension] denotes the feature dimensions. Aggregation indicates aggregation operation and MLP size.

| Module | npoint | [radii] | [nquery] | [dimension] | Aggregation |
|---|---|---|---|---|---|
| SA layer | 4096 | [0.2, 0.8] | [16, 32] | [[16, 16, 32], [32, 32, 64]] | MLP ($32 \rightarrow 64$) + MLP ($64 \rightarrow 64$) |
| SA layer | 1024 | [0.8, 1.6] | [16, 32] | [[64, 64, 128], [64, 96, 128]] | MLP ($128 \rightarrow 128$) + MLP ($128 \rightarrow 128$) |
| SA layer | 512 | [1.6, 4.8] | [16, 32] | [[128, 128, 256], [128, 256, 256]] | MLP ($256 \rightarrow 256$) + MLP ($256 \rightarrow 256$) |
| Vote layer | 256 | - | - | - | MLP ($256 \rightarrow 128 \rightarrow 3$) |
| SA layer | 256 | [4.8, 6.4] | [16, 32] | [[256,256,512], [256,512,1024]] | MLP ($512 \rightarrow 512$) + MLP ($1024 \rightarrow 512$) |

## B    DETAILED DETECTOR ARCHITECTURE

We report the detailed architecture of our DBQ-SSD. DBQ-SSD is a single-stage point-based detector that consists of three Set Abstraction (SA) layers for extracting point features, and one SA layer for aggregating centroid-based instances. Each SA layer has two different groups for the spherical neighbor query. In addition, a vote layer is used to generate candidate points. A light-weight head is attached to the backbone to predict final results. The detailed architecture for KITTI is reported in Tab. 4.

The head consists of two parallel branches, *i.e.*, classification and regression branches. The corresponding architecture:

$$Classification\ branch:\ FC(512) \rightarrow FC(256) \rightarrow FC(256) \rightarrow FC(3)$$

$$Regression\ branch:\ FC(512) \rightarrow FC(256) \rightarrow FC(256) \rightarrow FC(30)$$

where the classification branch predicts 3 classes, while the regression branch predicts 12 classes of equally angle bins and their corresponding angle offsets, and the distance $(d_x, d_y, d_z)$ to its corresponding instance, as well as the size $(d_l, d_w, d_h)$. For Waymo scene, we use the same model setting and just adjust the scale of input point cloud to 16,384, 4,096, 2,048 and 1,024.

## C    EXPERIMENTS ON TITTI *val* AND ONCE *val* SET

To verify the generalization, we evaluate our method on both KITTI *test* set and ONCE *val* set.

**KITTI *val* set.**    As shown in Tab. 5, our DBQ-SSD achieves comparable performance compared with IA-SSD, while showing super inference speed nearly two times than IA-SSD.

**ONCE *val* set.**    Because the official configuration file of IA-SSD is not released with respect to ONCE dataset, we reproduce the results according to the paper. As shown in Tab. 6, our method significantly improves the inference speed to **33 FPS (2.4x speedup)**, while maintaining comparable performance with IA-SSD. When adjusting the $\gamma$ to 0.1, our method achieves nearly **1 mAP** performance improvement while gaining **1.7x speedup**.

Table 5: Comparison with the state-of-the-art methods on the KITTI *val* set. The average precision is measured with 11 recall positions. The bold font is used to indicate the best performance. The speed is tested on a single GPU with a batch size of 16 and measured by FPS.

| Method | References | Type | Car Mod. (IoU=0.7) | Pedestrian Mod. (IoU=0.5) | Cyclist Mod. (IoU=0.5) | Speed (FPS) |
|---|---|---|---|---|---|---|
| VoxelNet (Zhou & Tuzel, 2018) | CVPR 2018 | 1-stage | 65.46 | 53.42 | 47.65 | 4.5 |
| SECOND (Yan et al., 2018) | SENSORS 2018 | 1-stage | 76.48 | - | - | 20 |
| PointPillars (Lang et al., 2019) | CVPR 2019 | 1-stage | 77.98 | - | - | 42.4 |
| TANet (Liu et al., 2020) | AAAI 2020 | 1-stage | 77.85 | 63.45 | 64.95 | 28.5 |
| Associate-3Ddet (Du et al., 2020) | CVPR 2020 | 1-stage | 79.17 | - | - | 20 |
| SA-SSD (He et al., 2020) | CVPR 2020 | 1-stage | 79.91 | - | - | 25 |
| CIA-SSD (Zheng et al., 2020) | AAAI 2021 | 1-stage | 79.81 | - | - | 32 |
| Part-$A^2$ (Shi et al., 2020b) | TPAMI 2020 | 2-stage | 79.47 | **63.84** | **73.07** | 12.5 |
| Fast Point R-CNN (Chen et al., 2019) | ICCV 2019 | 2-stage | 79.00 | - | - | 16.7 |
| STD (Yang et al., 2019) | ICCV 2019 | 2-stage | 79.80 | - | - | 12.5 |
| PV-RCNN (Shi et al., 2020a) | CVPR 2020 | 2-stage | **83.90** | - | - | 12.5 |
| VIC-Net (Jiang et al., 2021) | ICRA 2021 | 1-stage | 79.25 | - | - | 17 |
| PointRCNN (Shi et al., 2019) | CVPR 2019 | 2-stage | 78.63 | - | - | 10 |
| 3D IoU-Net (Li et al., 2020a) | Arxiv 2020 | 2-stage | 79.26 | - | - | 10 |
| Point-GNN (Shi & Rajkumar, 2020) | CVPR 2020 | 1-stage | 78.34 | - | - | 1.6 |
| 3DSSD (Yang et al., 2020b) | CVPR 2020 | 1-stage | 79.45 | - | - | 25 |
| IA-SSD (Zhang et al., 2022) | CVPR 2022 | 1-stage | 79.57 | 58.91 | 71.24 | 83 |
| Efficient Baseline | ICLR 2023 | 1-stage | 79.17 | 58.11 | 69.29 | 102 |
| DBQ-SSD | ICLR 2023 | 1-stage | 79.56 | 58.82 | 71.03 | **162** |

Table 6: Comparison with the state-of-the-art methods on the ONCE *val* set. Bold font is used to indicate the best performance. The speed is tested on a single GPU with batch size of 16 and measured by FPS.

| Method | Vehicle | | | | Pedestrian | | | | Cyclist | | | | mAP | Speed |
|---|---|---|---|---|---|---|---|---|---|---|---|---|---|---|
| | Overall | 0-30m | 30-50m | >50m | Overall | 0-30m | 30-50m | >50m | Overall | 0-30m | 30-50m | >50m | | |
| PointPollars | 68.57 | 80.86 | 62.07 | 47.04 | 17.63 | 19.74 | 15.15 | 10.23 | 46.81 | 58.33 | 40.32 | 25.86 | 44.34 | - |
| SECOND | 71.19 | 84.04 | 63.02 | 47.25 | 26.44 | 29.33 | 24.05 | 18.05 | 58.04 | 69.96 | 52.43 | 34.61 | 51.89 | - |
| PV-RCNN | **77.77** | **89.39** | **72.55** | **58.64** | 23.50 | 25.61 | 22.84 | 17.27 | 59.37 | 71.66 | 52.58 | 36.17 | 53.55 | - |
| PointRCNN | 52.09 | 74.45 | 40.89 | 16.81 | 4.28 | 6.17 | 2.40 | 0.91 | 29.84 | 46.03 | 20.94 | 5.46 | 28.74 | - |
| IA-SSD | 70.30 | 83.01 | 62.84 | 47.01 | **39.82** | **47.45** | 32.75 | 18.99 | 62.17 | 73.78 | 56.31 | **39.53** | 57.43 | 14 |
| IA-SSD (Reproduced) | 70.48 | 84.16 | 63.77 | 49.27 | 38.22 | 44.14 | 33.10 | 20.41 | 61.90 | 73.94 | 55.44 | 38.37 | 56.87 | 14 |
| DBQ-SSD ($\lambda$=0.05) | 72.06 | 84.63 | 64.66 | 50.13 | 38.32 | 43.35 | 32.97 | 21.22 | 62.16 | 73.94 | 56.65 | 38.20 | 57.51 | 23 |
| DBQ-SSD ($\lambda$=0.10) | 72.14 | 84.81 | 64.27 | 50.22 | 37.83 | 43.88 | 32.18 | 20.29 | **62.99** | **75.13** | 56.65 | 38.91 | **57.65** | 24 |
| DBQ-SSD ($\lambda$=0.20) | 71.63 | 84.38 | 64.06 | 49.82 | 37.27 | 41.90 | **33.59** | **20.95** | 62.77 | 74.94 | **57.14** | 38.47 | 57.22 | 27 |
| DBQ-SSD ($\lambda$=0.30) | 70.66 | 83.28 | 63.66 | 48.88 | 37.46 | 42.35 | 32.94 | 22.21 | 62.51 | 74.46 | 56.65 | 38.01 | 56.88 | **33** |

# D VISUALIZATION

As shown in Fig. 5, Fig. 6, and Fig. 7, we provide the detailed visualization of predicted results for Waymo *val* set and KITTI *val* set. The conclusion is the same as KITTI scene. As the network depth increases, the foreground points are retained for classification and regression, while redundant background points are dropped. It reveals that our method can adaptively discard useless points for speeding up inference. It's worth noting that the discarding behavior of point clouds significantly differs between KITTI and Waymo scenes, which verifies that our method equips generalization.

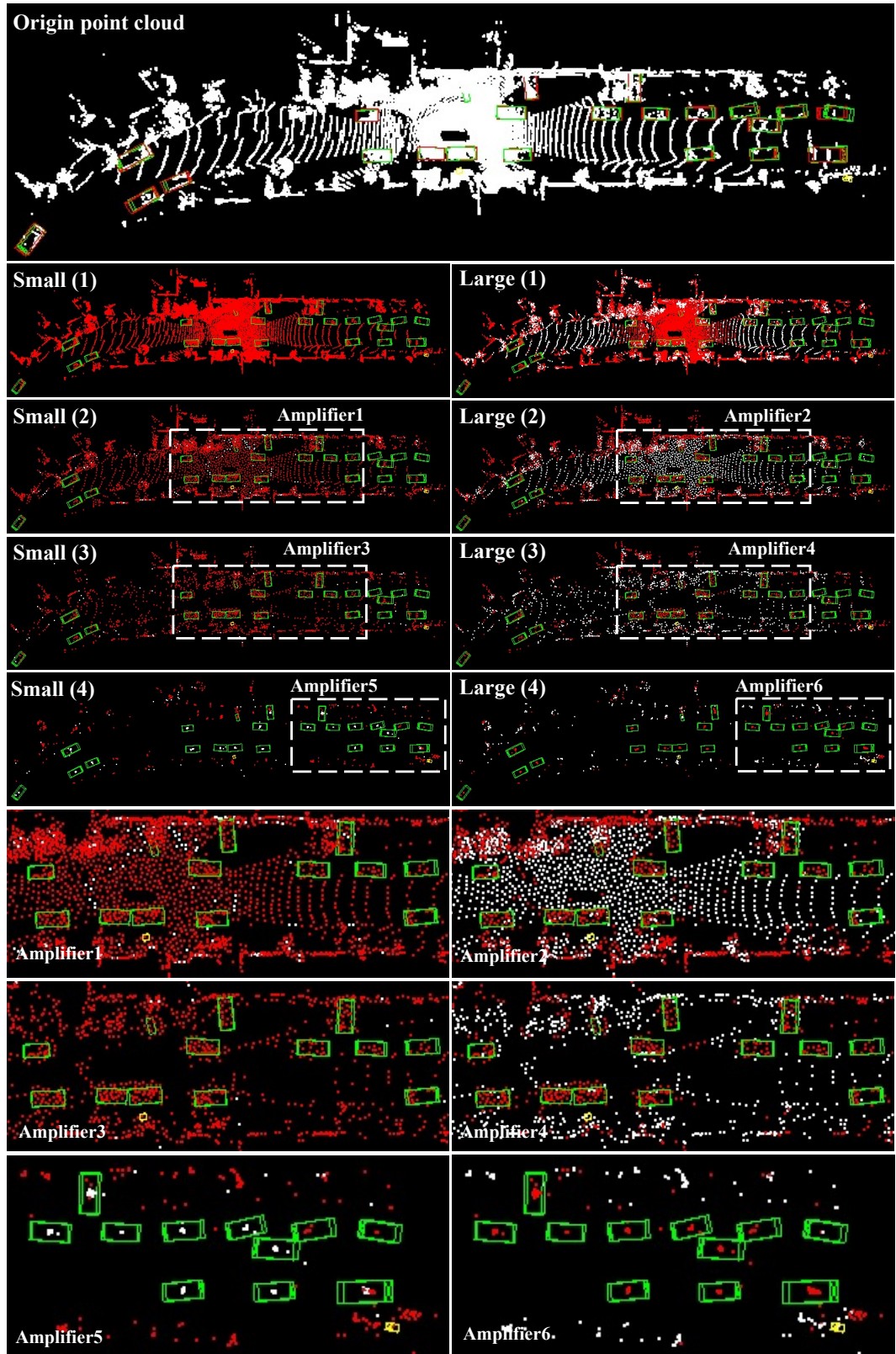

Figure 5: Visualization results on Waymo *val* set. The red and green 3D boxes in figures are ground truth and prediction boxes. Green, cyan, and yellow represent *Car, Pedestrian, and Cyclist*. Red and white points represent activation and blocking points, respectively. "Small" and "Large" means the scale of group in MSG, and the digital in parentheses is the index of SA layer.

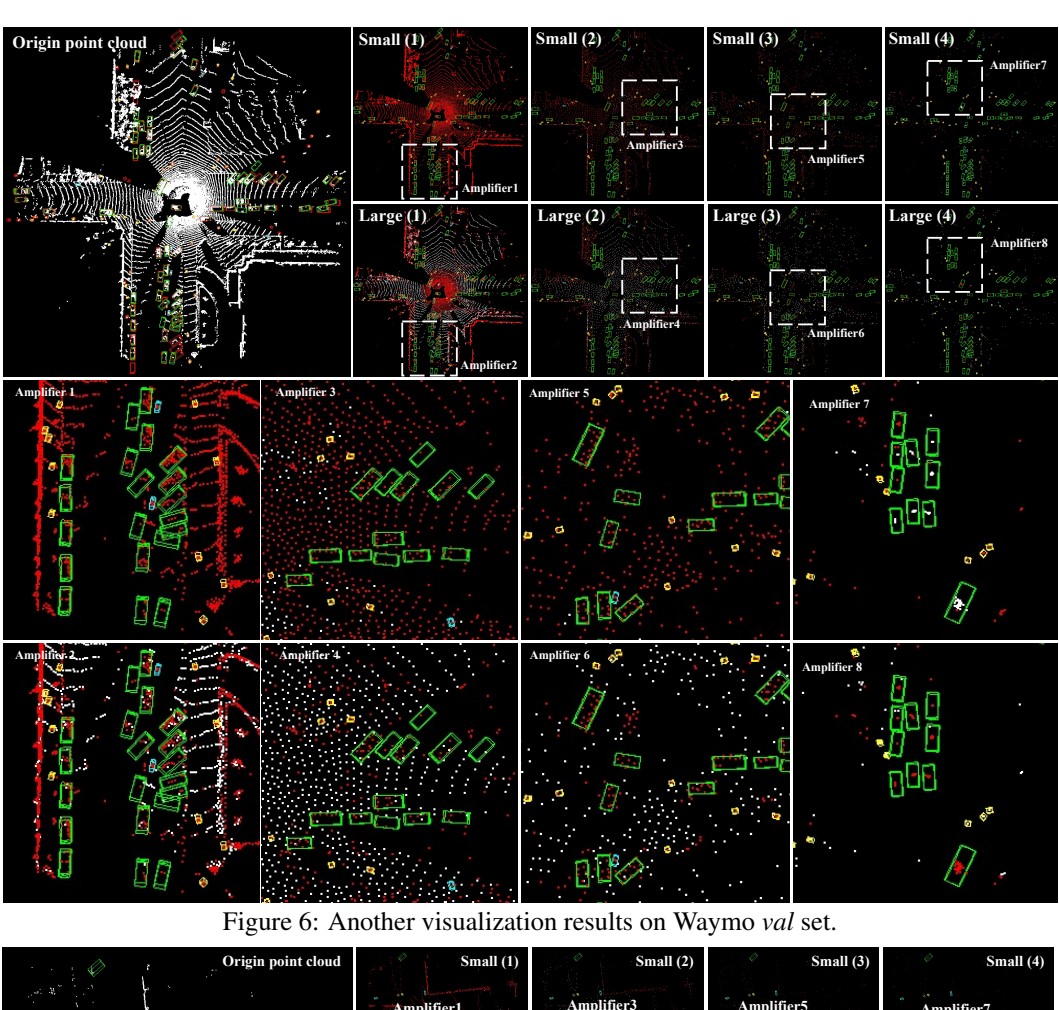

Figure 6: Another visualization results on Waymo *val* set.

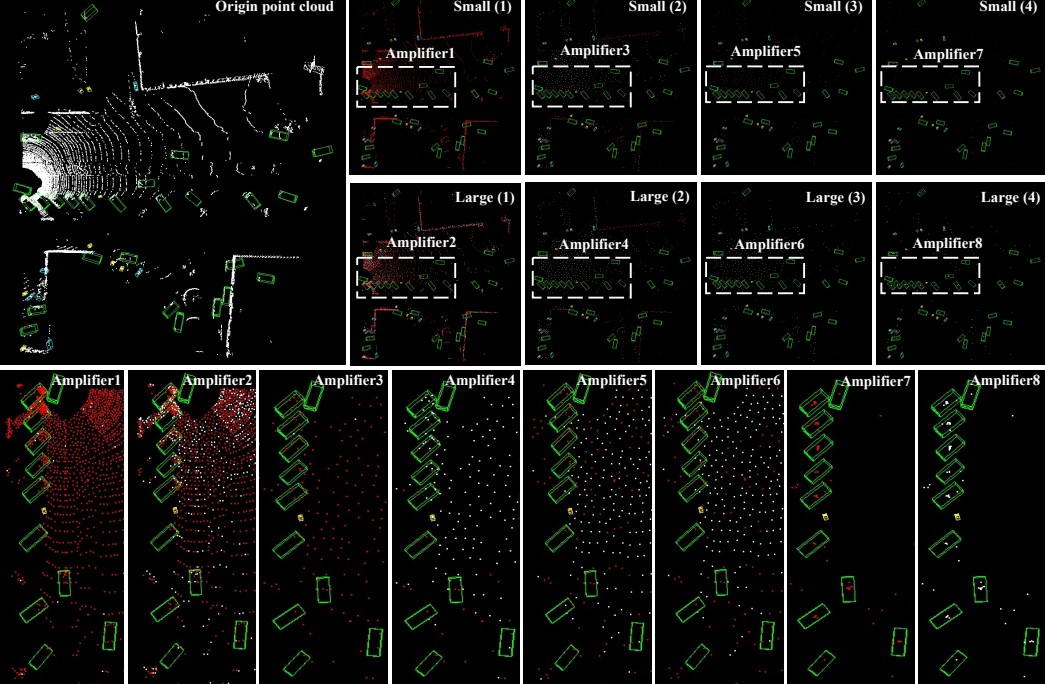

Figure 7: Detail visualization results on KITTI *val* set. The red and green 3D boxes in figures are ground truth and prediction boxes. Green, cyan, and yellow represent *Car, Pedestrian, and Cyclist*. Red and white points represent activation and blocking points, respectively. "Small" and "Large" means the scale of group in MSG, and the digital in parentheses is the index of SA layer.

