# OpenReview forum: "DBQ-SSD: Dynamic Ball Query for Efficient 3D Object Detection"
_ICLR.cc/2023/Conference — ICLR 2023 poster_

### Official Review · Reviewer_Bins · 2022-10-19

**Confidence:** 4
**Correctness:** 4
**Technical Novelty And Significance:** 3
**Empirical Novelty And Significance:** 3
**Recommendation:** 8

**Clarity, Quality, Novelty And Reproducibility:**

See above regarding quality, clarity, and originality.

Minor points:

- Abstract: "Reduce latency by 100%" - this is somewhat ambigious, does this mean the runtime drops to zero? Which whould be a rather bold claim.

- p.4: Multiple expressions are used for different kind of nearest neighbor searches, and it is not always clear which means what. What exactly is "nearest sampling technique", "top-k", "ball gathering", and "vanilla ball query". It would help if a formal definition of those would be given, or a more common name (such as "k-NN", epsilon-search), and / or a consistent name throughout the manuscript.
 Also, "For each query, the vanilla ball querying samples a predefined number of local points within a specific spatial radius." - isn't it *either* a kNN-query (select k nearest neighbors), OR an epsilon-search (select all points with distance < epsilon, i.e. a "specific spatial radius")?


**Strength And Weaknesses:**

- The overall **idea** is novel, well motivated and well thought off. Especially the parameterization of speed vs. accuracy using an additional loss is very interesting and allows tuning that trade-off without changing the architecture. The building block is generic can can likely be introduced into any network that uses ball-like queries on 3D (or 2D) data.

- The paper is **well written**, the method description is easy to follow, all steps are well motivated and the method can likely be reproduced.

- The **experiments** are thorough and fair. They are performed on large state-of-the-art datasets and the method is compared to many prior art methods. Additional experiments shed more insight into the model's inner workings and performance, further showing the internal impact of the proposed dynamic radii selection.

- The **results** are good and show that the proposed method improves the baseline and exceeds state of the art in the cost-vs-performance tradeoff.

**Summary Of The Paper:**

This manuscript proposes an extension to point-cloud processing networks that use nearest neighbor queries. Instead of fixing the search radii for those queris before training, the radii are selected dynamically on a per-point basis, and the selection criteria is learned. An additional loss is used to penelize the computational cost induced by selecting multiple radii. The experiments are well-designed and show that this approach significantly improves the cost-vs-performance tradeoff compared to prior art.

**Summary Of The Review:**

The paper presents a novel, well thought of and motivated idea that is fairly and thoroughly evaluated and exceeds state of the art on some metrics. Its general design has the potential to improve several 3D point cloud processing networks that use range queries.

---

> ### Author Response · Authors · 2022-11-16
> **Response to Reviewer Bins**
>
> We appreciate the reviewer's recognition of our work and insightful suggestions.
>
> **Q1: The ambiguous presentation in the abstract.**
>
> **A1:** Sorry for the confusion. Compared with the baseline IA-SSD, the inference speed of our work increases by up to 100%. We have improved the abstract in the revision and described the efficiency more precisely.
>
> **Q2: Confusion about the definition of multiple nearest neighbor searches.**
>
> **A2:** Thanks for the constructive suggestions. Accordingly, we added detailed definitions of different methods to the revision. Besides, vanilla ball querying refers to the ball querying proposed by PointNet++. It employs a k-epsilon-search strategy, which randomly samples k points in a predefined distance. We have also clarified it in the revision.

---

### Official Review · Reviewer_iTND · 2022-10-23

**Confidence:** 4
**Correctness:** 3
**Technical Novelty And Significance:** 3
**Empirical Novelty And Significance:** 3
**Recommendation:** 6

**Clarity, Quality, Novelty And Reproducibility:**

The motivation of this paper is clear. The idea of using a dynamic network to drop redundant background points is novel. The results may be hard to reproduce if the authors do not open the source code.

**Strength And Weaknesses:**

Strengths:

S1: The motivation of this paper is clearly presented. The paper analyzes the latency of existing point-based detectors and reveals the spatial redundancy existing in features of background points in Sec. 3. The observation may inspire studies of efficient point-based detectors.

S2: The idea of using a dynamic network to drop redundant background points is novel. In order to block redundant background points, the authors propose to predict the gating logits, which are used to select a subset of input queries for each group in MSG operation. Therefore, the method chooses optimal grouping branches to process point features and avoids useless receptive field groups.

S3: The experimental evaluations are comprehensive to support the claim. The proposed method presents an impressive improvement in inference speed over state-of-the-art and achieves comparable accuracy on KITTI, Waymo, and ONCE datasets.

Weakness:

W1: The authors claim the method achieves higher inference speed (162 vs 102). My concern is whether the setting of the baseline is reasonable enough. As shown in Fig. 3(d), no points go through the large-scale group in the fourth SA layer. The baseline is expected to achieve faster speed if it discards the useless grouping operation in SA layers.

W2: In Table. 1, the paper lists the performance of dynamic gating with different routing manners. What does the layer-wise routing mean? Please supplement a clear definition for this in the text.

W3: The visualization results on KITTI val set are not informative enough; please show local details and make an analysis about why most points are blocked at the large-scale group in fourth SA layer.

Typos:
1. Page 1, “INTRODUCATION” ⇒ “INTRODUCTION”

2. Page 1, “Yang et al. (2020b) first proposes a single stage architecture”⇒ “Yang et al. (2020b) first propose a single stage architecture”

3. Page 3, "The valuable observation motivates us to block and choose optimal grouping branches" => "The valuable observation motivates us to choose optimal grouping branches".

4. Page 6, “need to be averaged along”⇒ “needs to be averaged along”

5. Page 8, “point cloud which scale is” ⇒”point cloud whose scale is”

6. Page 9, “with with batch” ⇒ “with batch”

7. Page 9, “the inference speed of efficient the baseline”⇒ “the inference speed of the efficient baseline”

8. Page 9, “When conducting larger scale”⇒ “When conducting a larger scale”

9. Page 9, “which further deepen the understanding”⇒”which further deepens the understanding”


**Summary Of The Paper:**

This paper points out the spatial redundancy on background points and useless group operation in MSG for the inappropriate receptive field. The authors present an algorithm named dynamic ball query, which dynamically generates gate masks for each group of MSG to process useful points and block redundant background points. The method is validated on three detection datasets, where a noticeable improvement in inference speed is observed.

**Summary Of The Review:**

Unlike existing point-based detectors that focus on recalling more foreground points with efficient sampling policies, the main contribution of this paper is choosing optimal grouping branches to process point features. I appreciate the approach, but more abundant quantitative and qualitative results are necessary to demonstrate the effectiveness of the proposed method. On the other hand, the authors should correct the typos and clarify some statements.

---

> ### Author Response · Authors · 2022-11-16
> **Response to Reviewer iTND**
>
> **Q1: Is the setting of the baseline reasonable enough?**
>
> **A1:** This is a thoughtful question. We believe the baseline setting is selected reasonably. To address this concern, we conduct several ablation studies in Table 6 and Table 7, and give the empirical analysis as follows. All the large group mentioned below refer to the large group in the fourth SA layer.
>
> (1) The large group deprecation only occurs in the KITTI dataset. As shown in Table 6, in the Waymo and Once datasets, a large portion of the points is enabled.
>
> (2) For our dynamic network, although the large group is disabled during inference, it still affects the optimization during training and the inference accuracy. As shown in Table 7, the static-baseline network (B1 vs. A1) exhibits non-negligible performance degradation in the absence of the large group. However, our dynamic network (A3 vs. A1) can significantly improve the inference speed while maintaining similar accuracy.
>
> (3) Without the large group, our dynamic network can still achieve much higher inference speed and comparable performance. As shown in Table 7, our approach (B3 vs. B1) improves the inference speed by 53% over the efficient baseline.
>
> **Table 6:** Comparison of the proportion of enabled points in the fourth SA layer.
> |Dataset|Small|Large|Kill|Small & Large|
> |:-|:-:|:-:|:-:|:-:|
> |KITTI|100%|0%|0%|0%|
> |Waymo|69.3%|19.8%|3.2%|7.7%|
> |ONCE|60.5%|21.3%|5.6%|12.6%|
>
> **Table 7:** Comparison of different group settings of the fourth-layer on the KITTI val set. The speed is evaluated on an Nvidia Geforce 2080Ti GPU with a batch size of 16.
> ||Method|Group|3D Car (IoU=0.7)|3D Pedestrian (IoU=0.5)|3D Cyclist (IoU=0.5)|Speed (FPS)|
> |:-|:-|:-|:-|:-|:-|:-|
> |A1|baseline|small, large|79.57|58.91|71.24|83|
> |A2|efficient baseline|small, large|79.17|58.11|69.29|102|
> |A3|our($\lambda$=0.1)|small, large|79.56|58.82|71.03|162|
> |B1|baseline|small|79.19|57.38|70.56|90|
> |B2|efficient baseline|small|79.06|57.25|70.49|107|
> |B3|ours($\lambda$=0.1)|small|79.11|57.29|70.64|164|
>
> **Q2: What does layer-wise routing mean?**
>
> **A2:** Layer-wise routing refers to controlling an entire SA layer rather than individual points.  We conduct this ablation study to demonstrate the superiority of our point-wise routine manner.
> We have elaborate on it in the revision.
>
> **Q3: The visualization results on the KITTI val set are not informative enough.**
>
> **A3:** Thanks for the valuable suggestion. We have provided figures with more informative details in the revision of our supplementary.
>
> **Q4: Why are most points blocked at the large-scale group in the fourth SA layer?**
>
> **A4:** Please refer to the answer in A1. In addition, as a dynamic network, our dynamic ball query pursues a better tradeoff between effectiveness and efficiency. Compared with the three preceding layers, the fourth SA layer contains fewer points but more feature channels that bring much greater computational costs. By adaptively reducing the point number in the fourth SA layer, our method can significantly improve the inference speed and preserve more network branches simultaneously, making the detector maintain competitive accuracy.
>
> **Q5: Some typos.**
>
> **A5:** Thanks for the thorough review! We have corrected these typos in the revision according to the comments.

---

### Official Review · Reviewer_FVXp · 2022-10-24

**Confidence:** 4
**Clarity, Quality, Novelty And Reproducibility:** Has been discovered before.
**Correctness:** 4
**Technical Novelty And Significance:** 1
**Empirical Novelty And Significance:** 1
**Recommendation:** 5

**Strength And Weaknesses:**

However, the core contribution of this paper, the dynamic ball query, is not new. In Pyramid R-CNN[1], the authors proposed an almost the same algorithm. Both of these two papers use gumble-sigmoid trick to learn the radius, and they are all applied on the 3d objection detection task.
I believe that the authors haven't seen [1] before. But based on the existence of [1], I still vote to reject this paper.

[1] Mao, J., Niu, M., Bai, H., Liang, X., Xu, H., & Xu, C. (2021). Pyramid r-cnn: Towards better performance and adaptability for 3d object detection. In Proceedings of the IEEE/CVF International Conference on Computer Vision (pp. 2723-2732).

After reading the rebuttal, I still think that the idea of learning the radius of a ball query is not particularly novel, even though there are some differences in the implementation details and the position of the module. However, I missed another key point: the authors argue that the radius can be learned to reduce the latency of the network. Given this contribution, I am inclined to give a borderline recommendation.
Additionally, I noticed that the authors keep mentioning "latency" in the introduction and method sections, but only "FPS" is evaluated in the experiments. In an autonomous driving scenario, "latency" is a more important metric than "FPS", since point clouds are typically processed frame by frame with a batch size of 1. The authors should consider replacing all mentions of "FPS" with "latency" to better align with real-world applications.

**Summary Of The Paper:**

In this paper, the authors propose to learn the radius in ball query with gumble-sigmoid trick. Experiments on both KITTI and Waymo show the algorithm is more efficient and effective than previous state-of-the-art point-based 3d object detectors.

**Summary Of The Review:**

See above

---

> ### Author Response · Authors · 2022-11-16
> **Response to Reviewer FVXp**
>
> We appreciate the review's comments but believe there exist big misunderstandings. Our work has several significant differences with the mentioned Pyramid R-CNN framework in terms of objective, method, and experiment. We outline the primary distinction as follows. In addition, we have added a reference to the Pyramid R-CNN and provided a comparison in the revision.
>
> **D1: Different objectives**
>
>
> The primary objective of Pyramid R-CNN is to learn the radius to improve the **accuracy of two-stage detectors** instead of pursuing higher efficiency.
>
> Our work intends to improve the **efficiency of single-stage detectors** while maintaining competitive accuracy by adaptively selecting the network branches for each input point.
>
> **D2: Different architectures and methods**
>
> Pyramid R-CNN:
>
> (1) The Density-Aware Radius Prediction module learns the sampling radius for each region-of-interest (RoI), whose mechanism is more akin to a **deformable attention** than dynamic routing. In particular, the topological structure of the network **never changes** for distinct inputs during inference, which prevents the framework from increasing efficiency.
>
> (2) Because the Density-Aware Radius Prediction module is based on ROIs, only two-stage 3D detectors can use it directly.
>
> (3) As far as we know, the Pyramid R-CNN employs a re-parameterization trick to make the radius sampling differentiable but without Gumbel distribution.
>
> Ours:
>
> (1) The dynamic ball query is based on the **dynamic routing** mechanism, which can **adaptively change** the topological structure of the network. During inference, our framework can disable specific network branches for each input point to significantly increase processing speed. By the way, the branch can also be the Density-Aware Radius Prediction module in Pyramid R-CNN.
>
> (2) Our dynamic ball query is **more generic**, which can be embedded into various point-based detectors, including the Pyramid R-CNN. Additionally, our work allows each point to be routed into multiple branches simultaneously, such as multiple SA layers with different radii (small and large receptive fields).
>
> (3) Although the Gumbel-sigmoid is a well-known trick, we study and verify the effectiveness of making the dynamic routing learnable for 3D object detection.
>
> **D3: Different experiments and results.**
>
> Pyramid R-CNN:
>
> (1) Although the framework is verified to improve accuracy over the PV-RCNN baseline but brings **over 10% latency overhead**.
>
> (2) As a part of the framework, the Density-Aware Radius Prediction module has few ablation studies. Besides, it only achieves marginal accuracy gains over the baseline, e.g., from 75.26% mAP to 75.63% mAP on the Waymo dataset.
>
> Ours:
>
> (1) Our framework is about **20x faster** than the Pyramid R-CNN on a Nvidia V100 GPU. Even compared with IA-SSD, the state-of-the-art efficient detector, our method achieves **102% inference speed up** (168FPS vs. 83FPS) with competitive accuracy.
>
> (2) We provide comprehensive ablation studies and visualizations to show the effectiveness of our dynamic routing process, including the latency, activation ratio, and gating predictions in various layers. Besides, we evaluate our method on more datasets.

---

### Official Review · Reviewer_ATJW · 2022-10-24

**Confidence:** 4
**Correctness:** 3
**Technical Novelty And Significance:** 3
**Empirical Novelty And Significance:** 3
**Recommendation:** 6

**Clarity, Quality, Novelty And Reproducibility:**

The paper introduced an important issue of existing point-based object detection framework as the spatial redundancy problem. Through technical novelty in the individual parts are limited, the overall framework seem to be interesting and perform well in provided experiments. Most of description seems to clear and be detailed for reproducing the proposed method.

**Strength And Weaknesses:**

[+] First of all, the authors explained the motivation of the paper very clearly and convincingly through the analysis of the various aspects in the existing point-based object detection model. Motivated by routing process of dynamic network, the proposed ball querying module seems to be well designed to adaptively select a subset of input points as queries and extract local features in a suitable spatial factor. Lastly, the proposed model can be easily used into the existing point-based detection model framework to replace the set abstraction layer.

[-] One concern is how well routing procedure works. There seems to be a lack of detailed explanation about whether it works well if it is divided more finely than 2 groups (large and small), and what kind of points is when multiple selections are made.

For efficiency, the authors used the nearest sampling, but it seems that the density change in terms of the distance from the LiDAR sensor does not take into account. Object nearby sensor has many points and farthest objects have few points, so nearest sampling can collect points that are completely unrelated to the farthest object.


**Summary Of The Paper:**

In this paper, the authors pointed out the issue of the spatial redundancy of existing point-based object detection frameworks. To handle this issue, the authors proposed the dynamic ball querying module, which is adaptively activated or blocked with a suitable combination among pre-defined group set. With this module and a resource budge loss, the model can be trained for achieving a better balance between effectiveness and efficiency. In experimental section, the authors provided various comparison to demonstrate that the proposed method can reduce effectively latency by a large margin on popular point cloud detection benchmarks.

**Summary Of The Review:**

I mentioned all comments including reasons and suggestions in the above sections. I recommend that the author will provide all the concerns, and improve the completeness of the paper.

---

> ### Author Response · Authors · 2022-11-16
> **Response to Reviewer ATJW**
>
> **Q1: How well does the routing procedure work when dividing more finely than two groups (large and small)?**
>
> **A1:** We appreciate the valuable suggestion. Accordingly, we conduct fine-grained experiments by adding a medium group to each SA layer. More components introduce more static costs of branch switching, making the baseline method (B1 vs. A1) less efficient. Nevertheless, our dynamic ball query can improve accuracy while reducing a larger proportion of the computation cost. As reported in Table 4, our method with three groups (B4) can slightly increase accuracy while cutting more than half of the latency. Specifically, it improves by 117% and 91% speed over the baseline (B1) and the efficient baseline (B2), respectively.
>
> **Table 4:** Comparison with different group settings on the KITTI validation set. The speed is evaluated on an Nvidia Geforce 2080Ti GPU with a batch size of 16.
> |Index|Method|Group|3D Car (IoU=0.7)|3D Pedestrian (IoU=0.5)|3D Cyclist (IoU=0.5)|Speed (FPS)|
> |:-|:-|:-|:-|:-|:-|:-|
> |A1|baseline|small, large|79.57|58.91|71.24|83|
> |A2|efficient baseline|small, large|79.17|58.11|69.29|102|
> |A3|ours($\lambda$=0.1)|small, large|79.56|58.82|71.03|162|
> |B1|baseline|small, medium, large|79.32|58.64|72.34|60|
> |B2|efficient baseline|small, medium, large|79.21|58.37|72.11|68|
> |B3|ours($\lambda$=0.1)|small, medium, large|79.63|58.89|72.60|110|
> |B4|ours($\lambda$=0.2)|small, medium, large|79.54|58.58|72.66|130|
>
> **Q2: What kind of points is enabled when multiple selections are made?**
>
> **A2:** As shown in Table 5, we divide all the 3D objects in KITTI val set into three categories according to the object size. The statistics demonstrate that large-size objects prefer larger groups in SA layers to capture long-range dependencies. In comparison, small-size objects also use smaller groups to obtain local relations and detailed structures for high effectiveness.
>
> **Table 5:** The proportion of enabled points in different groups of different layers on KITTI val set. 'S', 'M' and 'L' indicate small object, medium object and large object. 'L1', 'L2', 'L3' and 'L4' refer to four consecutive SA layers, respectively.
> |Group|L1(S)|L2(S)|L3(S)|L4(S)|L1(M)|L2(M)|L3(M)|L4(M)|L1(L)|L2(L)|L3(L)|L4(L)|
> |:-|:-:|:-:|:-:|:-:|:-:|:-:|:-:|:-:|:-:|:-:|:-:|:-:|
> |small| 100%|98.3%|99.6%|97.4%|   100%|89.0%|95.7%|0.3%|    100%|78.9%|92.8%|0.6%|
> |medium|0.4%|71.8%|2.3%|100%|     1.7%|12.8%|54.6%|100%|    0.5%|7.8%|64.4%|100%|
> |large| 100%|95.7%|0%|0%|           100%|92.1%|0%|0%|         100%|85.9%|0%|0%|
>
> **Q3: Why use the nearest sampling for the multiplexer feature?**
>
> **A3:** As the reviewer mentioned, we chose the nearest sampling due to its high efficiency. However,  the nearest sampling could be replaced by numerous alternatives to increase effectiveness even more. We believe this is an open question that deserves further study in future work.

---

### Decision · Program_Chairs · 2023-01-20

**Decision:**

Accept: poster

**Justification For Why Not Higher Score:**

This paper receives 2x marginally above the acceptance threshold and 1x strong reject.

**Justification For Why Not Lower Score:**

This paper has 1x accept, good paper.

**Metareview: Summary, Strengths And Weaknesses:**

This paper receives 2x marginally above the acceptance threshold, 1x accept, good paper, and 1x strong reject.

All the three reviewers who give accepts agree that the authors explained the motivation of the paper very clearly and convincingly through the analysis of the various aspects in the existing point-based object detection model. The idea of using a dynamic network to drop redundant background points is novel. The experimental evaluations are comprehensive to support the claim. The experiments are thorough and fair. The results are good and show that the proposed method improves the baseline and exceeds state of the art in the cost-vs-performance tradeoff. The weaknesses mentioned by these reviewers are mostly additional experimental results and further clarifications that are addressed in the authors responses.

The metareviewer carefully considers the 1x strong reject, where the main reason is the proposed method is similar to pyramid RCNN. Both papers propose the ball query method. However, the authors' response managed to clarify the issue. Specifically, the authors the clarifications that they have different objectives, propose different architectures and methods, and show different experiments and results compared to pyramid RCNN. The metareviewer is convinced that this paper is different from pyramid RCNN.

**Note From Pc:**

if the above contains the word "oral" or "spotlight" please see: "oral" presentation means -> notable-top-5% and "spotlight" means -> notable-top-25%. As stated in our emails, we are disassociating presentation type from AC recommendations